# Kinesin Motors in the Filamentous Basidiomycetes in Light of the *Schizophyllum commune* Genome

**DOI:** 10.3390/jof8030294

**Published:** 2022-03-12

**Authors:** Marjatta Raudaskoski

**Affiliations:** Molecular Plant Biology, Department of Life Technologies, University of Turku, 20014 Turku, Finland; marrau@utu.fi

**Keywords:** filamentous fungi, cytoskeleton, microtubules, motor molecules, intracellular transport, nuclear division

## Abstract

Kinesins are essential motor molecules of the microtubule cytoskeleton. All eukaryotic organisms have several genes encoding kinesin proteins, which are necessary for various cell biological functions. During the vegetative growth of filamentous basidiomycetes, the apical cells of long leading hyphae have microtubules extending toward the tip. The reciprocal exchange and migration of nuclei between haploid hyphae at mating is also dependent on cytoskeletal structures, including the microtubules and their motor molecules. In dikaryotic hyphae, resulting from a compatible mating, the nuclear location, synchronous nuclear division, and extensive nuclear separation at telophase are microtubule-dependent processes that involve unidentified molecular motors. The genome of *Schizophyllum commune* is analyzed as an example of a species belonging to the Basidiomycota subclass, Agaricomycetes. In this subclass, the investigation of cell biology is restricted to a few species. Instead, the whole genome sequences of several species are now available. The analyses of the mating type genes and the genes necessary for fruiting body formation or wood degrading enzymes in several genomes of Agaricomycetes have shown that they are controlled by comparable systems. This supports the idea that the genes regulating the cell biological process in a model fungus, such as the genes encoding kinesin motor molecules, are also functional in other filamentous Agaricomycetes.

## 1. Introduction

*Schizophyllum commune* Fr. is a model fungus for the genetic, molecular, and cell biological studies of filamentous basidiomycetes. The classical genetic studies revealed that the sexual reproduction in filamentous basidiomycetes is regulated by incompatibility factors *A* and *B* [1]. *B* regulates fertilization with the reciprocal nuclear exchange and migration of nuclei between the mates with different *B* factors, while the A factor regulates the pairing of the nuclei with different *A*s and *B*s. The incompatibility factors are now named *A* and *B* mating-type genes, and their structure is well-known. Mating-type *B* genes encode a pheromone receptor, as well as pheromones, while A consists of homeodomain genes [2,3]. These structures of mating-type genes have been confirmed today in several filamentous basidiomycetes [4,5]. The activity of mating-type genes leads to the establishment of a dikaryon from homokaryons with different *A* and *B* genes. The mating and growth of dikaryotic hyphae consists of extensive inter- and intracellular nuclear movements, respectively, but the genetic background and cell biology of these phenomena are still poorly known.

Early cell biological studies have shown that the microtubule cytoskeleton is involved both in the mating and growth of the haploid and dikaryotic hyphae of filamentous basidiomycetes [6] and, in *S. commune*, the depolymerization of microtubules affects the apical cell morphology and stops the growth [7,8]. Recently, visualizing living homokaryotic and dikaryotic hyphae with fluoroprotein-labelled nuclei and actin have shown an actin cytoskeleton in association with the early phase of nuclear division both in homokaryotic and dikaryotic hyphae [9,10]. At telophase, nuclear division is associated with fast nuclear movement [9]. According to our indirect immunofluorescence microscopical studies, the nuclear movement at early mitosis and at telophase follows microtubule tracks [11,12] and probably requires microtubule-associated motor molecules. At matings with different *B* mating-type genes, cytoplasmic and spindle microtubules occur in the hyphal bridges and in the hyphae involved in fusions [4,13]. All these observations indicate an important role for the microtubule cytoskeleton and call for a study of the motors involved in the processes. It starts here by thr analyses of *S. commune* kinesins in the light of the known functions of animal, yeast, and ascomycete kinesins.

### 1.1. The Number of Kinesins in Eukaryotic Organisms

A eukaryotic organism typically has two types of microtubule-associated motor molecules, one (or a few) dynein motors, and several kinesin motors. One recent study has characterized the dynein motor protein and its function in *S. commune* hyphae [14]. Before the characterization in *S. commune*, the two-partite structure of dynein was only known from the smut fungus *Ustilago maydis* DeCandolle Corda among basidiomycetes [15]. In the budding yeast *Saccharomyces cerevisiae* Meyen ex E.C. Hansen five [16], in the fission yeast *Schizosaccaromyces pombe* Lindner nine [17,18], in the filamentous ascomycetes *Aspergillus nidulans* (Eidam) Winter and *Neurospora crassa* Shear et Dodge [18,19], and in the basidiomycete smut *U. maydis* [20], nine to eleven kinesin motor proteins have been identified, respectively, but an extensive survey of the kinesins found in filamentous basidiomycetes (Agaricomycetes) is missing.

The number of kinesins is low in fungi, while humans have 45 known kinesins and Arabidopsis has 61 known kinesins [21]. Today, kinesins are divided into 14 families [22,23]. In the present review, as in most recent reports, the nomenclature and division of Lawrence et al. [22] is followed, although in all eukaryotic organisms the original name of an identified kinesin gene or protein is still in use, but often with a mention of which established kinesin family the investigated kinesin belongs to. In most organisms, including yeasts and filamentous ascomycetes, a kinesin family contains members developed through the duplication and divergence of the encoding gene. In the budding yeast, the five kinesin genes belong to four kinesin families [16]. *S. pombe* kinesin-8 and -14 families [17], and in *A. nidulans* kinesin-3 family, have each two members [19]. 

### 1.2. Kinesin Structure and Function

Kinesin proteins consist of four domains: a globular catalytic kinesin motor, followed by the neck, coiled-coil stalk, and the tail. The globular motor consists of eight beta sheets surrounded, on both sides, by three alpha helixes and loops between these structures.The motor part is responsible for the movement of the protein along the microtubules. In the motor, the conserved amino acid sequences are involved in the binding and hydrolysis of ATP and in maintaining and releasing ADP and P_i._ This region is known as the P-loop, with the sequence GxxxxGKT. Switch I and Switch II are involved in the conformational changes of the motor structure, depending on ADP/ATP binding [24]. The part that binds to β-tubulin in microtubules takes place at the terminal part of the motor domain [25]. The kinesin motors are mainly active as homodimers, but also as heterodimers. Enzymatic activity has to be coordinated between the two motor heads so that one of the kinesin motors is always attached to the microtubule, while the other unit hydrolyses ATP, releases P_i_, and is separated from the microtubule [24].

An unbound kinesin motor is loaded with ADP, but binding to tubulin leads to the release of ADP from the active site, followed by the binding of ATP and a switch in the motor structure [26]. The release of P_i_ at the hydrolysis of ATP brings the motor molecule one step forward on the microtubule track and detaches it from the microtubule [24,27,28]. The energy released by the rear motor of the dimer flips it over the front motor and separates it from the microtubule. The release of ADP from the kinesin motor that is now in front binds it back to the microtubule, and ATP hydrolysis, along with flip over movement in the rear motor, may take place.

On the microtubule, the kinesin motors are always 8 nm apart. With the hydrolyses of ATP and the flip over the front motor, the rear motor steps 16 nm from its previous site [29,30,31]. These hand-over-hand steps may take place on the same microtubule 200 times, bringing the kinesin molecule over two microns forward. The motor is, therefore, processive. In addition to the regulation of the processivity, the kinesin motors influence microtubule polymerization, stability, and depolymerization. Most kinesins have their motor domain at the N-terminus of the polypeptide, and these N-kinesins move toward the plus end of the microtubule. In a few kinesins, the motor part is in the C-terminus of the protein, and these kinesins are moving towards the minus end of the microtubule. There are also a few kinesins known from animal cells, where the motor is in the middle of the molecule. These M-kinesins regulate the depolymerization of microtubules [32]. 

The hand-over-hand step, or the rear motor flip over the front motor, is regulated by the neck region in the homo- and heterodimeric kinesins [33,34,35]. The neck linker consists of 14 to 18 amino acids, extending from the last alpha helix (alpha 6) of the motor domain to the first coiled-coil dimerization of the stalk (helix 7) [35,36]. In spite of the short length, the configuration changes in the amino acid chain of the neck linker regulate and co-ordinate the hand-over-hand movement of the kinesin motor domains in the dimer. The neck linker peptide has a strictly ordered structure due to the binding to specific amino acids in the motor domain when the kinesin motor binds to ATP, but it becomes relaxed as a consequence of ATP hydrolysis, which facilitates the forward step on the microtubule [36,37]. 

The dimerization of a kinesin is regulated by the first coiled-coil structure in the neck region. The first amino acids of the coiled-coil domain form hydrophobic interactions between the two chains and, thus, they regulate the stabilization of the dimerization of the kinesin motor pair [34]. Behind the neck linker, the structure of the stalk varies in different kinesins in terms of the number and distribution of coiled-coils and the uncoiled regions, hinges between them [38]. In kinesin-1 in animal cells, the kinesin light chains (KLC) bind to the terminal part of the last coiled-coil part in the stalk. In fungal kinesins, no kinesin light chains have been identified [38].

The stalk domain is followed by the tail domain, which may consist of two separate linear chains or of a randomly coiled part. The tail functions in cargo transport, interacting with different signaling molecules, and in the autoinhibition in the absence of cargo [38]. The latter phenomenon is known in kinesin-1, -2, -3, and -7. The autoinhibition has been studied extensively in kinesin-1, which exists in a folded autoinhibited conformation in vivo. During autoinhibition, the stalk assumes a folded conformation, which is helped by the hinge region in the stalk and by the attachment of a specific amino acid sequence, IAK, of the tail to the motor domain switch I [39,40]. This attachment hinders ADP release from the motor induced by the motor binding to a microtubule. In animal kinesin-1, the kinesin light chains (KCLs) are also involved in autoinhibition by contributing to the maintenance of the folded state of the stalk, as well as by pushing apart the motor domains [41]. The autoinhibition is released by post-translational modifications, such as phosphorylation, cargo binding, or by specific proteins [38]. It is noteworthy that while the motor domain structure of all the known kinesins is conservative, the different stalk and tail structures identify and regulate the various functions of kinesins. 

### 1.3. Schizophyllum Commune Kinesins

Analyzing the *S. commune* genome (Schco3/Schco3.home.html; accessed on 7 February 2022) with the kinesin-1 motor domain from human, rat, fission yeast, and filamentous ascomycetes *Aspergillus nidulans* (kinA XP_662947.1), *Neurospora crassa* (Nckin, >EAA35196.2), and smut basidiomycete *Ustilago maydis* (Ukin2, U92845) identifies ten kinesin proteins in the genome (Figure 1). Further comparisons and characterizations of the amino acid sequences of the neck linker, stalk, and tail show that the ten proteins belong to nine kinesin families, with kinesin-7 including two members (Figure 1). In all identified kinesin motor domains, the sequences of the P-loop, Switch I, Switch II, and microtubule binding regions are present with only small variations. 

The alignment of the motor domains of *S. commune* and of two other filamentous basidiomycetes, *Coprinopsis cinerea* (Schaeff.) Redhead, Vilgalys et Moncalvo, and *Laccaria bicolor* (Maire) Orton, with well-known motor domains of yeast and filamentous ascomycetes, confirm the family placement or grouping of the filamentous basidiomycete kinesins (Figure 2, Table 1). Seven of the fungal kinesins are involved in mitosis, although some of them may also have functions not directly involved in mitosis. Currently, no association with mitosis is reported for kinesin-1 and -3. These two kinesins are discussed first. 

## 2. Cytoplasmic Transport Kinesins

### 2.1. Kinesin-1, Schcokin1

The kinesin-1 family proteins are N-terminal kinesins, which move towards the plus end of the microtubules. The first kinesin-1 protein was identified by Vale et al. [72] as a translocator along microtubules in squid axoplasm. In filamentous fungi, kinesin-1 was cloned and sequenced ten years later in ascomycete *Neurospora crassa* and was named Nkin, but it was also known as conventional kinesin or kinesin heavy chain (KHC) [73]. Shortly thereafter, two kinesin genes were cloned, one of them kin1, in the plant pathogenic basidiomycete *Ustilago maydis* [74]. In *N*. *crassa* studies, it was indicated that fungal kin1 was a dimer without kinesin light chains (KLC). The purified glass attached motor of *N. crassa* moved in in vitro experiment microtubules with a high speed of 2.5 µm s^−^^1^. This is three times faster than for animal kin1, whose speed is 0.6 µm s^−1^ [75,76]. The properties of *N. crassa* kin1 led to the crystallization of the Nkin motor domain and its comparison with the crystal structure of rat kinesin RnKin [77,78]. However, no dramatic changes were recorded in the amino acid sequence or the crystal structure of the motor domain, which could clearly explain the high speed of the Nckin motor, apart from the more efficient ATP hydrolysis [78].

For the *Schizopyllum commune* kinesin-1 protein (PID2605236, in Schco3/Schco3.home.html; accessed on 7 February 2022), there are three amino acid sequences in GenBank: the identical ACF7313/ACG58879.1 sequences, and the XP_003036134.1 sequence, consisting of 969 and 981 amino acids, respectively. The ACG58879.1/ACG58879.1 amino acid sequence is based on the isolation of kinesin 1 clones from the genome and the cDNA libraries of *S. commune*. The protein XP_003036134 contains an extra intron removed from PID202605236 (Figure 1), but not from the GenBank sequence XP_003036134. The Schcokin1 N-terminal motor domain runs from amino acid five to 332 (InterProScan, Figure 1B). The comparison of the motor part and the whole kinesin sequences, with the corresponding sequences of *N. crassa* Nckin (EAA35196.2) and *U. maydis* Kin1 (U92845), shows 69% identity and 80% similarity in the motor domains and 51% identity and 66% similarity in the whole proteins.

In both fungal and animal kinesins, the neck domain, consisting of a neck-linker and the first coiled-coil part in the stalk, has an important role in regulating movement, direction, and velocity. It also coordinates the function of the two motor domains in the dimeric kinesins [33,34,35,36,37]. In Schcokin1, the 14 amino acids of the neck-linker, from amino acid 332 to 346 after the motor domain, are highly homologous with the neck linker in *U. maydis* and *N. crassa*. However, the amino acids in the coiled-coil part belonging to the neck domain are more variable, though this part starts with proline in all three fungi [35]. When the coiled-coil part of the neck region is run with the coiled-coil programs [93], a strong coiled-coil signal is obtained in *U. maydis* kin1, while in Nckin1, the signal is weaker, and in Schcokin1, it is barely detectible. In the coiled-coil part of the *N. crassa* kin1 neck domain, tyrosine 362 (Y362) has been extensively studied. Its mutations reduce the stability of the dimerization of the coiled-coil part, but they also affect the catalytic activity of the motor domain and its velocity [94,95,96,97]. Y362 is also conserved in the coiled-coil part of the neck domain in Schcokin1 and *U. maydis* kin1, but no detailed studies on the function of the neck domain are yet available in basidiomycetes.

The stalk part of Schcokin1 starts with a 65 amino acid-long hinge, followed by about 100 amino acids in coiled-coil 1. The 70 amino acid kink region separates coiled-coil 1 from the long, almost 300 amino acid-long coiled-coil terminal part, which can be divided into coiled-coil 2 and the tail (Figure 1B). Interestingly, the coiled-coil tail part is highly conserved in fungal and animal kinesins, and also in Schcokin1. The sequence functions in cargo binding [98]. The tail end of Schcokin1 also contains the short motif IAK that regulates the autoinhibition of all conventional kinesins.

The deletion mutants of fungal kinesin-1 are not lethal, as in animal cells [58,74,99]. In *N. crassa* Nckin, *A. nidulans* kinA, and *U. maydis* Umkin1 (kin2), the deletion of the *kin1* gene strongly reduced the hyphal growth rate, and the electron microscopic examination of the mutant strains showed deficiencies in the “Spitzenkörper” structure [74], which is the vesicle-distributing center for polarized hyphal growth. Currently, experiments have confirmed that KinA in *A. nidulans* and UMkin1 in *U. maydis* are involved in the transport of chitin synthases and vesicles packed with cell wall synthases along the microtubules toward the hyphal tip [59,81]. In the living hyphae of the Nckin1 deletion mutant, the GFP-labeled vesicles showed disturbances in the assembly into the Spitzenkörper [71], while in *A. nidulans*, KinA is shown to cooperate with myosin-5 in transferring secretory vesicles from the microtubules into the actin network, during exocytosis, at the hyphal tip [79].

In *N. crassa* and *A. nidulans* multinucleate hyphae, the disruption of kinesin-1 genes caused the aggregation of nuclei, and the distance between the nuclei and the hyphal tip increased [58,99]. The same phenomenon was also seen in *N. crassa* living hyphae with fluorescent protein-labelled nuclei [71]. Some small disturbances in nuclear movements and positions were also recorded in *U. maydis* dikaryotic hyphae, with both nuclei carrying the deletion of *Umkin1* gene [74]. The research in *A. nidulans* [60] indicated that the effect of the kinesin-1 gene deletion on the nuclear aggregation is indirect. In the absence of the KinA protein, the dynein localization to the microtubule plus ends is impaired, which affects dynein-related nuclear migration. The effect of fungal kinesin-1 gene disruption on nuclear distribution is of great interest in *S. commune* and in other filamentous basidiomycetes with extensive intracellular nuclear migration in association with nuclear division.

### 2.2. Kinesin3, Schcokin3 

In addition to kinesin-1, kinesin-3 is also involved in intracellular transport. Schcokin3 (PID 2276863) is a large protein consisting of 1597 amino acids, with an N-terminal motor between amino acids 6-376, followed by the FHA domain from amino acid 490 to 592 and a plekstrin domain at the C-terminus from amino acid 1496 to 1590 [100] (Figure 1, InterProScan). Very few coiled-coil structures are predicted, only the dimerization domain after the motor domain between amino acids 400 and 500 and in the stalk domain around amino acids 700 to 900. Kinesin-3 was described in the early nineties in *C. elegans* as UNC104 [100] and in the murine brain as KIF1A [101]. The high homology of these proteins indicated that they belonged to the same kinesin family. KIF1A was described as a unique monomeric, globular fast motor protein with the subdomains described above. After the early work, it has been demonstrated that kinesin-3 proteins show concentration-dependent dimerization [38].

In the early work [101], KIF1A was shown to be involved in the transport of membrane organelles, associated with specific synaptic vesicle proteins and Rab3A. In *U. maydis*, Kin3 is also involved in the transport of small vesicles named early endosomes (EE). In yeast-like sporidia, the transport was observed to take place along microtubules in a cell-cycle dependent manner [82,83]. The deletion mutant of *Umkin3* was not lethal but it reduced the particle movement by 33%, while the depolymerization of microtubules completely abolished the movement of EE particles. The remaining motion of particles in the *Umkin3* deletion strain was considered to be due to the minus-end motor protein dynein, which moves the early endosomes towards the minus end of microtubules [82,83].

The relationship between the two motors, dynein and kinesin-3, in the bidirectional transport of endosomes has also been studied in the growing hyphae of *U. maydis*. The recording of the distribution of the plus and minus ends of microtubules in hyphae showed that the plus ends of the microtubules are oriented toward the hyphal tip and septa, while the antiparallel microtubules overlap in the central part of the tip cell [84]. Observing long-distance transport in a live hypha with motors labelled with fluorescent proteins indicated that Umkin3 is always attached to endosomes, and it carries endosomes toward the tip. Endosomes with Umkin3 are also captured by retrograde-moving dynein, present close to the tip, and are transported toward the central part of the hypha. In the region with the antiparallel microtubules, endosomes with Umkin3 are released from dynein, and the direction of movement changes toward the plus ends of the microtubules, towards the tip or septum. The accumulation of dynein at the hyphal poles, at the tip, and at the septum, ensures efficient loading onto dynein, keeping endosomes on their tracks [85,86,87,88]. The bidirectional endosomal movement could be necessary for communication between the extending tip and the nucleus, or formation of the septum at the end of the tip cell. Kinesin-1 is involved in the traffic by transporting dynein to the plus ends of the microtubules close to the hyphal tip and septum [84].

Kinesin-3 has also been cloned and characterized in *A. nidulans*, UncA [61], and in *N. crassa*, Nkin2, [80]. Both in *A. nidulans* and *N. crassa*, there are shorter homologues to kinesin-3, UncB, and Nkin3, respectively, each consisting of a motor domain and a small part of the stalk. In *U. maydis*, a second kinesin-3 protein was not detected, nor was it detected in the *S*. *commune* genome. In both *A. nidulans*, UncA, and *N. crassa*, Nkin2, the GFP/mRFP-labelled Kin3 was localized to fast-moving vesicles along the hypha. The colocalization with endosomal FM4-64 and Rab5a-labelled vesicles indicated that Kin3 was associated with bidirectionally moving endosomes. In both fungi, the deletion of Kin3 caused a reduction in hyphal polarized growth, and the movement of vesicles ceased during microtubule depolymerization. These data indicated that kinesin-3, in filamentous ascomycetes, is also involved in vesicle transport. In *A. nidulans*, UncA appeared to follow microtubule tracks consisting of detyrosinated alpha-tubulin [61], and in *N. crassa* as well, the movement of Kin3 appeared to be associated with microtubules consisting of a specific type of tubulin [80]. No such association with specific microtubules was detected in *U. maydis* [89].

An exciting view has been opened by demonstrating that the microtubule cytoskeleton and kinesins are needed for the long-distance transport of mRNA along microtubules to enable fast polar growth in *U. maydis* hyphae [89,102,103,104]. The UmKin3–dynein endosomal transport mechanism has been revealed to be responsible for the bidirectional transport of the mRNA-containing particles in the hyphal tip cells, while the role of kinesin-1 is the distribution of dynein [90]. The plekstrin domain at the C-terminal of kinesin-3 plays a central role in vesicle movement. In addition to mRNA, polysomes are also transported on the endosomes, allowing protein synthesis to take place during the transportation process. Lipid droplets, peroxysomes and the endoplasmic reticulum may also comigrate with the endosomes [91]. Recently, the core components of endosomal mRNA transport have been identified in the basidiomycetes *Coprinopsis cinerea* and *Amanita muscaria* [105]. This suggests that the mRNA transport, during the hyphal tip growth of filamentous basidiomycetes, could be a process comparable to the one in *U. maydis* infectious hyphae.

## 3. Mitotic Kinesins

The best-known fungal mitotic kinesins are kinesin-5 and -14. These mitotic kinesins are discussed first and then according to kinesin family number.

### 3.1. Kinesin-5, Schcokin5

The function of kinesin-5 in mitosis was first reported in *A. nidulans* based on the genetic study of the *bimC*4 ts mutant [62]. Thereafter, kinesin-5 has been identified in all eukaryotic cells; in humans [106], in plants [107] and yeasts. Kinesin-5 is identified as Cut7 in *S. pombe* [51], and as Cin8 and Kip1 in *S. cerevisiae* [42,108]. In *S*. *commune*, kinesin-5 (Schcokin5) consists of 1167 amino acids identified both in the *Schizophyllum* genomic (kinesin 2, EU880234) and the cDNA libraries (kinesin 2, EU850808.1), as well as in the whole genome sequence (PID 2605230, Figure 1A). The comparison of the genomic and cDNA sequences (EU880234 and EU850808) suggests that Schkin5 contains four introns, and three of them are close to the C-terminus. In EU850808.1, amino acids 338 and 339 are RR, but in PID 2605230, they are correctly AE.

According to InterProScan, the Schcokin5 motor domain extends from amino acid 55 to amino acid 428. Schcokin5 contains an extension of 55 amino acids in front of the motor domain (Figure 1B). The slightly longer N-extension in *S. pombe* Cut7 [106], and in *A. nidulans* Bimc [63], is suggested to enhance microtubule binding. The motor domain of Schcokin5 is highly homologous with Eg5, BimC, Cut7, Cin8, and Kip1 with 52%, 56%, 56%, and 40% identities, respectively, except for a sequence of 20 amino acids between the β5 strand and the α3 helix. The 20 amino acid sequence is close to, but not overlapping with, the long P8 loop typical to the budding yeast Cin8 [64]. The 20 amino acid insert is also present in the kinesin-5 motor domain of the basidiomycetes *U. maydis* and *C. cinerea*, which have highly identical motor domains with Schcokin5, with a 62% and 79% identity, respectively. In the Cryo-EM structure of the *U. maydis* kinesin-5 motor, the insert is named β5/Loop8 and is suggested to be involved in microtubule binding [92].

Behind the motor domain, in the N-terminal part of the Schcokin5 stalk, between amino acids 436 to 456, a coiled-coil region extends with leucine heptad repeats. This coiled-coil region is probably involved in the dimerization of the monomers, similar to the other kinesin-5 motors [43]. The coiled-coil regions 466–535, 643–663, and 762–789 at the stalk end of Schcokin5 could be involved in the formation of the BASS bipolar assembly domain from the two dimeric kinesin-5 molecules, leading to the establishment of the functional bipolar homotetramer that is typical to kinesin-5 proteins [109,110]. The tail of Schcokin5 includes the nuclear localization signal (NLS) and the BimC box typical to the kinesin-5 tail region, containing a consensus sequence for Cdk1 (Cdc2) phosphorylation [106].

*A. nidulans* BimC was discovered due to a mutation that blocked the beginning of mitosis by inhibiting the separation of spindle pole bodies [62]. Mutations in the *Cut7* gene of *S. pombe* and in *S. cerevisiae Cin8* and *Kip1* have a similar effect [42,51,108]. The kinesin-5 homotetramer, with a pair of plus-end-directed motors at both ends, crosslinks and slides apart the antiparallel microtubules leading to the separation of the spindle pole bodies and to the elongation of the spindle, especially at anaphase B [44,111]. During fungal mitosis, several other kinesins, including kinesin-14 [44,112], are involved. Recently, it was shown that *S. pombe* mitosis takes place when Cut7 and kinesin-14 are deleted. A small spindle is formed with the help of microtubule antiparallel bundler PRC1/Ase1, which recruits CLASP/Cls1 to stabilize microtubules. The power for the elongation of the spindle is due to the polymerization of the stable microtubules [113]. These components probably also play a role during normal mitosis. Although kinesin-5 homotetramers are plus-end motors, as individual molecules, the yeast kinesin-5 motors, Cin8 and Kip1, may function as minus-end motors with faster minus end, than plus end, motility [45,64]. *S. pombe* Cut 7 is able to function as a minus-end motor, both as a monomer, but also as a homotetramer, both in vitro and in vivo [52,114].

In the *S. commune* genome, the *Schcokin1* and *Schcokin5* genes are located only 960 bp apart and are translated on different strands and in a different direction. In the genomes of basidiomycetes *Coprinopsis cinerea* and *Laccaria bicolor*, the location of *kinesin-1* and -*5* genes is similar (Figure 3). Whether the close location is meaningful for the expression of the *kinesin-1* and -*5* genes remains to be clarified.

### 3.2. Kinesin-14, Schco14

In fungi kinesin-14 was detected by the identification of the gene *KAR3*, necessary for nuclear fusion in budding yeast *S. crevisiae* [50]. The gene encoded a kinesin-related protein, which was already known from Drosophila as the NCD protein with in vitro minus-end-directed motility [115]. With information from these two proteins (genes), a kinesin-14 encoding gene, KlpA, from *A. nidulans* was cloned and identified [69]. The *KlpA* deletion, by itself, had no phenotype, but it suppressed the effect of the bimC4 (kinesin-5 mutant allele) phenotype of *A. nidulans*, thus allowing spindle formation. From the very beginning, this led to the idea that kinesin-5 (BimC) and kinesin-14 (KlpA) are kinesin motors that have opposite effects during spindle formation [70]. Presently, kinesin-14 is known from several eukaryotic organisms and fungi, including fission yeast *S*. *pombe*, which have two kinesin-14 genes, *Pkl1* and *Klp2* [57], while budding yeast *S. cerevisiae* only has one kinesin14-gene, *Kar3*, but its product associates with two non-motor proteins, Vik1 and Cik1 [116].

In *S. commune*, the Schcokin14 protein (PID 2617151) consists of 773 amino acids and a C-terminal positioned motor domain between amino acids 349 and 771 (InterProScan, Figure 1B) suggesting microtubule minus-end-directed motility. A strong nuclear localization signal is predicted at the N-terminal tail from amino acid 11 to amino acid 38. The N-terminal stalk contains a coiled-coil structure between amino acids 245 and 272, and 301 and 348. The Schcokin14 neck-motor junction is similar to the one in the well-studied kinesin-14 motor proteins Ncd and Kar3 [117,118]. The last coiled-coil ends the amino acids 347G and 348N in front of the motor domain, as in the Kar3 and Ncd neck motor junction. The Schcokin14 protein shows between 40–50% similarity with the kinesin-14 proteins of filamentous basidiomycetes in the GenBank. The low identity/similarity is due to the fact that most of the kinesin-14 proteins reported in the GenBank are incomplete, including the one (XP_003032452.1) reported as kinesin-14 for *S. commune*. The protein is identical to PID 2617151 but lacks 184 amino acids from the N-terminus.

The Schcokin14 protein (PID2617151)-encoding gene is located on the plus strand of scaffold_4 between bps 133413-136762. Upstream on the plus strand, there are three additional genes encoding the amino acids of the kinesin-14 N-terminal part, first PID2746089 (151 amino acids, bps 125507-126062), then PID2534380 (329 amino acids, bps 116894-118166) and finally PID2676636 (414 amino acids, bps 108382-109795). One of these N-terminal kin-14 protein fragments (PID2534380) is also reported in the GenBank (XP_003032451.1). The three genes encoding N-terminal fragments of kinesin-14 (PID2676636, PID2534380 and PID2746089) probably result from the rearrangement of scaffold_4. The gene downstream from PID2676636 (bps 108382-109795) encoding the shank protein (PID2497768) also occurs next to the genes encoding kinesin fragments PID2534380 and PID2746089. The alignments of the nucleotides encoding the three fragments of Schcokin14 and the genes next to them show high homology. It could be that *S. commune* has two kinesin-14 encoding genes, like *S. pombe*, but the arrangements in scaffold_4 in front of the full-length *Schcokin14* gene may have destroyed the structure of the other kinesin-14 gene. Whether the 28000 bp structure of scaffold_4, including the *Schcokin14* gene, the three upstream N-terminal fragments, and the triplication of the *Shank* gene, is only typical to the *S. commune* strain H4-8 genome, and has to be investigated in the future.

### 3.3. Kinesin-4, Schcokin4

In screening the cDNA library of haploid and mated compatible haploid strains of *S. commune*, transcripts encoding the kinesin ACF75332.1 (kin3) with sequence homology to Kif4 from kinesin-4, the chromokinesin family, was identified as frequently as transcripts for Schcokin1 and -5. Schcokin4 (kin3) contains 1978, but the Schco_3 PID2563303 sequence contains 2000 amino acids, due to the presence of an extra intron in the motor domain. The motor domain of Schcokin4 extends from amino acid 6 to 429 and the stalk consists of several coiled-coil areas, probably necessary for homodimerization. In animal cells, KIF4 moves along microtubules toward the plus end, and a motility assay moves microtubules at a velocity of 0.2 μm/s [119]. This kinesin has also been reported from *A. nidulans* (An6875) [19] and *U. maydis* [20], and in the GenBank Blast, the Schcokin4 sequence aligns with numerous kinesin-4 proteins from filamentous basidiomycetes. Kinesin-4 from the filamentous basidiomycete *C. cinereus* shares a 53% identity and a 66% similarity with Schcokin4. 

Kif4A is necessary for chromosome condensation, spindle organization, chromosome alignment, and cytokinesis [120]. In the NCBIBlast (Pfam) search, the stalk area of Schcokin4 represents overlapping bacterial and eukaryotic SMC motifs involved in the structural maintenance of chromosomes, cell cycle control, cell division, and chromosome partitioning, supporting the idea that Schcokin4 is a chromokinesin involved in mitosis, although no nuclear localization signal was detected. Chromokinesin Kif4 is also necessary for DNA replication after mitosis and chromatin build-up [121]. It is intriguing to speculate that the high expression level of Schcokin4 in haploid and mated mycelium could not only be due to its presence at the beginning of mitosis at chromosome condensation, but also in the nuclei after mitosis. In *S. commune*, the size of the nuclei increases significantly during rapid telophase movement, suggesting the decondensation of chromatin associated with DNA replication and new chromatid synthesis [10]. All these functions are known to involve Kif4 in interphase human cells [121,122].

Kif4 has a well characterized role in cell division, but there is growing evidence that Kif4 has roles in non-dividing cells. KIF4, as a motor protein, is involved in the transport of specific substances in nerve cells and the regulation of the growth and apoptosis of nerve cells [123,124]. KIF4 also participates in the selective stabilization of cytoplasmic microtubules to assist cell migration in fibroblasts [124].

### 3.4. Kinesin-6, Rab6- Kinesin, Schcokin6

In fungi, kinesin-6 has been previously reported in *U. maydis* [20], with only a weak but significant similarity with the other representatives of the Kinesin-6 family. Recently, kinesin-6, known in *S. pombe* as Klp9, has been shown to be a homotetrameric, processive, plus-end-directed motor [53] with a role in mitosis at anaphase B, together with kinesin-5 and Map protein Ase 1 [53,125], in adjusting spindle elongation velocity to cell size [126]. Schcokin6 has a 19% identity and a 31% similarity with Klp9, and a 29% identity and a 44% similarity with *U. maydis* kinesin-6. It consists of 1029 amino acids (PID2564434), the N-terminus is made up of 119 amino acids localized internally, the stalk domain contains three small coiled-coil regions, and according to the NCBIBlast (pfam) analysis, there is, after the motor region, a Rab6-binding domain (RBD) at amino acids 696–738. A similar motif is recognized in human kinesin-6 proteins Kif20A and Kif20B, which is why Kinesin-6 is also named rab6-kinesin. Human kinesin-6 sequences, like Kif20A, KiF20B, have NLS and SMC regions, which suggests that they are involved in nuclear division, similar to *S. pombe* Kpl9, while the RBD domain is needed for cytokinesis [127,128]. Schcokin6 has two putative nuclear localization signals between amino acids 904–933 and 984–989 (NLS Stradamus). The NCBIBlast shows that kinesin-6 is highly detected in filamentous basidiomycetes. The role of the RBD motif is still unclear, as is the role of kinesin-6 in nuclear division in filamentous basidiomycetes. It would be of interest to investigate whether Schcokin6 might be a regulator of the anaphase B elongation rate as Klp9 is in *S. pombe*, and whether it has a role in the tight association between nuclear division and septa formation, in mono- and dikaryotic hyphae [20].

### 3.5. Kinesin 7, Schcokin7A, Schcokin7B

In the *S. commune* genome, there are two genes, one encoding 791 (PID2520682, 7A) and the other encoding 709 (PID 2522273, 7B) amino acids, with a sequence identity to the motor domain of the kinesin-7 family proteins. In the InterProScan analyses, Schokin7A is recognized with a homology to the motor domain CENP-E, and the shorter protein 7B with a homology to *S. cerevisiae* kinesin-7 protein Kip2p. Human CENP-P is large, with over 2000 amino acids, and is a dimeric protein consisting of an N-terminal globular motor domain, a long coiled-coil stalk, and a farnesylated tail [129]. It is a kinetochore-targeted motor that walks towards the microtubule plus ends. CENP-E, together with chromokinesin-4 and -10, is involved in the congression of chromosomes on the metaphase plate by leading and attaching the microtubules to kinetochores, and in this process the long stalk is important. *S. cerevisiae* Kip2p is a much smaller protein, consisting of 706 amino acids [130]. In fungi, kinesins belonging to family 7 have been investigated in *U. maydis* [20], in *S. pombe* Tea2p [54], and in *A. nidulans* KipA [65]. *U. maydis* kinesin 7a consists of 1465, and 7b of 1043, amino acids, while the Tea2p and KipA, proteins are smaller, consisting of 628 and 889 amino acids, respectively.

According to InterProScan, the N-terminal motor domain of Schcokin7A extends from amino acid 36 to amino acid 419, and in 7B from 1 to 334. The motor part of the *S. commune* proteins 7A and 7B shows a 33% and a 26% identity, and a 47% and a 39% similarity with the human CENP-E motor, respectively, but the stalk part is much shorter. The N-terminal part of both proteins might be longer, which will have to be reinvestigated in the future. Schcokin7A and 7B are 71% identical and 73% similar, with most of the dissimilarities existing in the motor part, which could be due to sequencing errors and the lack of cDNA information. The stalks of 7A, from amino acid 439 to 791, and in 7B from 361 to 706, are 92% identical and 95% similar, with only a few amino acid differences. In both proteins, coiled-coil regions are predicted, and in 7A, the first one just behind the motor domain between amino acids 420 and 447, and in the other one, between amino acids 505 and 536, suggesting that the 7A kinesin probably forms a dimer. In the stalk of 7B, the coiled-coils are predicted at the same sites. No nuclear localization signals are detected in the proteins. In sequence comparisons with *U. maydis* kin7 proteins, the *S. pombe* Tea2p and *A. nidulans* KipA identities and similarities for the total amino acid sequences are about 20% and 30%, respectively, and for the motor domain, they are about 10% higher. However, a more exact annotation is required for Schcokin7A and 7B.

The deletion of Tea2p and KipA causes changes in *S. pombe* cell morphology [54] and in the hyphal growth pattern of *A. nidulans* [65]. In *A. nidulans*, the deletion leads to curved hyphae due to the shift of the hyphal growth center, Spitzenkörper, from its central location in the hyphal tip [65,131]. The kinesin-7 motor in *S. pombe* and *A. nidulans* is shown to move processively towards the plus-ends of microtubules, and attach the microtubules at the tip cell cortex [132]. The partners interacting with Tea 2p and KipA at the tip have been identified [54,65,131,132]. In *A. nidulans*, the research inspired by KipA has led to completely new ideas about cytoskeletal interactions during hyphal tip growth [133]. In *A. nidulans*, KipA also occurs in association with the spindle of a dividing nucleus, where it was shown to bind the microtubule plus ends to a kinetochore protein with homology to CENP-H [66]. In animal cells, CENP-H is involved in binding CENP-E to kinetochores.

In *S. cerevisiae*, Kip2p regulates the length of the cytoplasmic microtubules, needed for the proper location of the nucleus during division [130] or for the separation of the spindle pole bodies [46]. Yeast Kip2p interacts with Bik1, a homolog to the microtubule, plus the end-binding protein, Clip-170, in mammals. The Kp2 and Bik1 protein complex associates with the minus-end-directed motor dynein and transports dynein to the plus ends of the microtubules for the interaction with the cargo for minus end transportation [47,48]. Thus, Kip2p plays an important role in the dynein transport to the microtubule plus ends, a function performed in filamentous fungi by kinesin-1 [59]. However, in *A. nidulans*, at an elevated temperature, KipA was found to enhance the accumulation of the ClipA/*A. nidulans* homolog to Clip-170, at the microtubule plus ends, while at normal growth temperature, KinA was responsible for the ClipA accumulation and its effects on microtubule dynamics and dynein transport [67]. This suggests that in filamentous fungi, kinesin motors responsible for central transportations could be able to respond to physiological conditions.

In *U. maydis*, the deletion of one or both kinesin 7 proteins had no effect on cell morphology, and the function of the proteins is unknown, perhaps more closely related to animal CENP-E. The kinesins 7A and 7B require attention in filamentous basidiomycetes, such as *S*. *commune*. Whether Schcokin 7A and 7B are encoded by two paralogous genes, or they represent two genes specified in functions, is of interest in filamentous basidiomycetes.

### 3.6. Kinesin-8, Schcokin8

The blast of the *S. commune* kinesin-like protein PID 2540501 with SwissProteins indicates that the motor region of the protein is about 50% identical with *S. pombe* Klp5 and Klp6, 47% identical with *S. cerevisiae* Kip3, and 47 to 48% identical with mouse Kif18A and KIf18B, which are all motor molecules in the kinesin-8 family. In yeasts [49,55], and in the filamentous ascomycete *A. nidulans*, KipB, [19], kinesin-8 members have been extensively investigated. In *S. pombe*, the early studies already indicated that kinesins Klp5 and Klp6 form heterodimers, which have a central role in regulating microtubule dynamics in metaphase and early anaphase [68]. The deletion of the kinesin-8 leads to the hyper-elongation of microtubules in *S. cerevisiae* and *S. pombe*, and, in humans, to the relaxation of spindle microtubules [56]. The dimeric structure of Kinesin-8 moves along all spindle microtubules towards their plus ends, where it is involved in either stabilization or depolymerization, which are processes important in the regulation of spindle microtubule lengths [134]. The effect of Kinesin-8 is not restricted to spindle microtubules, but it also controls the length of astral and cytoplasmic microtubules in all organisms studied. In *A. nidulans*, the deletion of the KipB encoding gene caused a delay in mitotic progression, and the mislocated, bent mitotic spindles. In wt hyphae, KipB localized towards the cytoplasmic, astral, and mitotic microtubules, and the green spots moved toward microtubule plus ends [19].

Schcokin8 consists of 1027 amino acids with a motor domain from amino acid 6 to amino acid 379, followed by coiled-coil regions 395–450, 431–451, and 556–585 (InterProScan), suggesting the dimerization of Schcokin8. The high identity (similarity) of the Schcokin8 motor, with the yeast and human motor regions, and the presence of the coiled-coil regions behind the motor, suggests that it could be a highly processive dimeric motor, like KIf18A or Kip3. The stalk and tail parts of Schocokin8 are quite different from yeast and human kinesin-8, but both parts contain several regions between amino acids 845 and 973, which predicts the nuclear localization of Schcokin8 by PSORTII and NLS, Stradamus. The tail of *S. cerevisiae* Kip3 and human Kif18A has an extra C-terminal microtubule-binding domain [135,136]. Whether the Schocokin8 tail part includes such a domain remains to be shown in the future.

### 3.7. Kinesin-10, Schcokin10

The *S. commune* genome contains a gene encoding a kinesin protein of 732 amino acids (PID 2671089) with a motor domain from amino acid 3 to amino acid 335 (InterProScan). In the SwissProtein Blast, the protein shows a 39% identity and a 60% similarity to the mouse and human kinesins Kif22/KID [137] and a 19% and a 33% to *Drosphila melanogaster* meiotic kinesin NOD [138]. These kinesins belong to the kinesin-10 family chromokinesins, which are able to bind to chromatin. The high similarity in the motor domain suggests that this *S. commune* kinesin is also a chromokinesin of the kinesin-10 family. Kinesin-10 chromokinesins are located in the nucleus and their function has been investigated in cells with large spindles, such as *Drosophila melanogaster* S2-cells [139] or human dividing HeLa cells [140]. Kinesin-10 members push the chromosomes from the poles toward the metaphase plate for chromosome congression, and the bidirectional attachment of kinetochores to opposite spindle poles. The phenomenon is called PEF (polar ejection forces). The NOD and KID kinesins are suggested to cause PEF in different ways. NOD is thought to be a nonmotile kinesin, binding chromosomes to microtubule plus ends, while KID is a plus-end directed dimeric motor, which transports the chromosomes along the microtubules to the metaphase plate [141].

The stalk and tail structure of *S. commune* kin-10 is different from KID and NOD. The Schcokin10 stalk is a coiled-coil from amino acid 470 to amino acid 497, and from 532 to 552. The latter coiled-coil structure is followed by a TPR (tetratricopeptide repeat) motif from amino acid 566 to amino acid 599, which is reported as a heptad wheel structure in the coiled-coil prediction. These regions could play a role in Schokin10 dimerization. The C-terminus between amino acids 627 and 730 includes several nuclear localization signals, proving that Schkin-10 is a nuclear kinesin, and the amino acid sequence PKKKRRRS predicts that the region also binds to DNA [142]. In the future, it is possible to confirm the localization of Schokin10 in the nucleus, but its function in mitosis will be difficult to demonstrate due to the small size or the near absence of the fungal metaphase plates. Few reports exist of Kinesin-10 (Kid) in filamentous fungi, but it is known from *A. nidulans* [19] (CBF75533.1 TPA), which shares a 25% identity and a 42% similarity with Schcokin10.

## 4. Are Kinesins Necessary for Growth and Nuclear Movements in Filamentous Basidiomycetes?

The high expression of Schcokin1, -3, -4, -5, and -14 in the preliminary experiments in the haploid and mated hyphae ([14], Raudaskoski unpublished experiments) sparks interest in their possible functions during vegetative growth and sexual reproduction. In the tip cells of *U. maydis* and *A. nidulans* [61,88], kinesin-3 is required for endosome and mRNA transport [143] toward the microtubule plus ends. In this system, dynein returns the vesicles with kinesin-3 to the central part of the hypha, while dynein transport to the hyphal tip takes place using the kinesin-1 motor. This is facilitated by the plus and minus orientation of the microtubules, with plus ends towards the hyphal tip and septum, but in an antiparallel orientation in the central part of the hypha [65,84]. This transport system is now well characterized, and it is probably present in filamentous basidiomycetes, which have an extensive microtubule cytoskeleton in the hyphal tip cells. All the three motors for the endosomal vesicle transport and the components of mRNA transport present in the genome [105], but this needs to be scrutinized. The investigation of filamentous basidiomycetes with a slightly different Spitzenkörper and closed septa in monokaryotic, multinucleate homokaryotic, and heterokaryotic, as well as dikaryotic, hyphae, and with septal dissolution during intercellular nuclear exchange and migration, could reveal new features in vesicle transport by kinesins in filamentous basidiomycetes.

Out of the ten kinesins identified or shown to exist in the *S. commune* genome, and probably in all other filamentous basidiomycetes, seven belong to mitotic kinesins. In five out of these seven kinesins, a nuclear localization signal is identified, indicating a role in the nucleus/mitosis. In animal cells [120], as well in fungi [54,65,68], the mitotic kinesins (kinesin-4, -7 and -8) are also active on the cytoplasmic microtubules. This is of interest with respect to the changes in the nuclear structure and movements, which takes place in association with nuclear division in homokaryotic and dikaryotic hyphae. At early mitosis, the nuclear size before division decreases significantly and then increases during the telophase movement, which is visualized recently in living hyphae [9,10], but was also known in the early days of cell biological studies of filamentous basidiomycetes [144]. Whether any of the mitotic kinesins have a role in the regulation of the nuclear size before and after mitosis will have to be investigated. The fast nuclear movement at telophase in the dikaryotic and monokaryotic hyphae of *S. commune* can be first visualized as an extensive anaphase spindle elongation, that is contributed to by astral microtubules interacting with hyphal cytoplasmic microtubules. The movement could be a result of the cooperation of several motors, including both kinesins and dynein. It is worth noting that several heterokaryotic filamentous basidiomycetes present a special location of the dividing nuclei to maintain a proper distribution of mating type genes [6], but the molecular background is not known.

In ascomycetes, dynein appears to be the most critical factor for nuclear distribution, and the dynein-mediated pulling force is also important for spindle orientation [145], as well as during the intercellular nuclear migration of *S. commune* [14]. However, in the latter case, the deletion of the heavy chain of dynein was not lethal and the haploid strains continued to exhibit slow growth. The high expression of *Schcokin5* and *Schcokin14* in the mutant strain was thought to explain the viability of the deletion mutant [14] suggesting that kinesins might also play a role in nuclear movements in vegetative hyphae. At mating, the mutant strain was not able to accept nuclei from the wild-type mate or to develop into a dikaryotic mycelium. Only unilateral nuclear migration from the mutant strain into the wild-type strain took place. This suggests that intercellular nuclear migration is highly dependent on dynein, but in growing hyphae, dynein may be replaced by other molecular motors, including kinesins.

## 5. Summary and Prospects

The present review presents the kinesins of *S. commune*, defining the motor regions and some generally known motifs in the stalk and tail. In several mitotic kinesins, a second microtubule binding site, not dependent on ATP hydrolysis, is well known [21]. It is outside the motor and is necessary for function, but not presented here. The functions of kinesins are also dependent on the association with specific proteins, and often regulated post-translationally by phosphorylation and dephosphorylation. This is not covered here, but it is still important for kinesin functions. The saprotrophic filamentous basidiomycetes grow fast and extensively on different substrates, and the nuclear division run by mitotic kinesins is a part of this rapid extension of growth. Kinesins also transport building materials for the hyphal extension and enzymes working on extracellular nutrient sources. It is interesting to see how the processes involving kinesins are modified in plant pathogenic or ectomycorrhizal filamentous basidiomycetes. A challenge for the future is to reveal the role of kinesins and dynein in the intra- and intercellular nuclear movements unique to filamentous basidiomycetes. The nuclear movements have been described for decades [146,147,148,149] without knowing the molecular mechanisms behind them.

## Figures and Tables

**Figure 1 jof-08-00294-f001:**
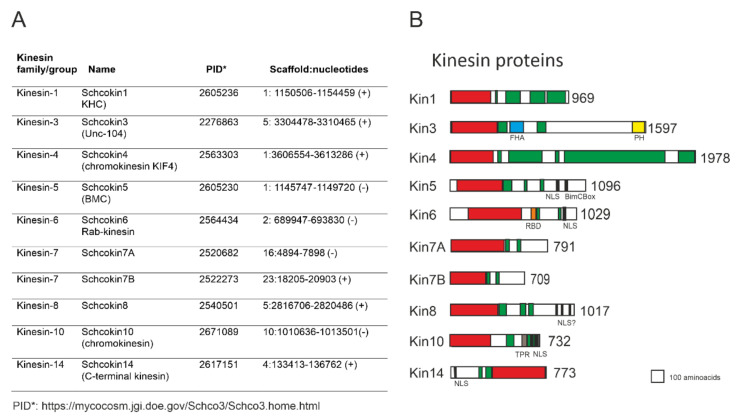
*Schizophyllum commune* kinesin genes and proteins. (**A**) The location of the kinesin genes in the *S. commune* genome. (**B**) A schematic drawing of kinesins with various domains and amino acid counts. The motor domain is red, the FHA forkhead-associated domain is blue, the PH plekstrin homology domain is yellow, the RBD Rab6-biding domain is orange, the TPR tetratricopeptide repeat domain is dark yellow (InterProScan and NCBIBlast), the coiled-coils are green (InterProteinScan) and the NLS nuclear localization signal bar is black (NLStradamus/localization signal, NLS Mapper).

**Figure 2 jof-08-00294-f002:**
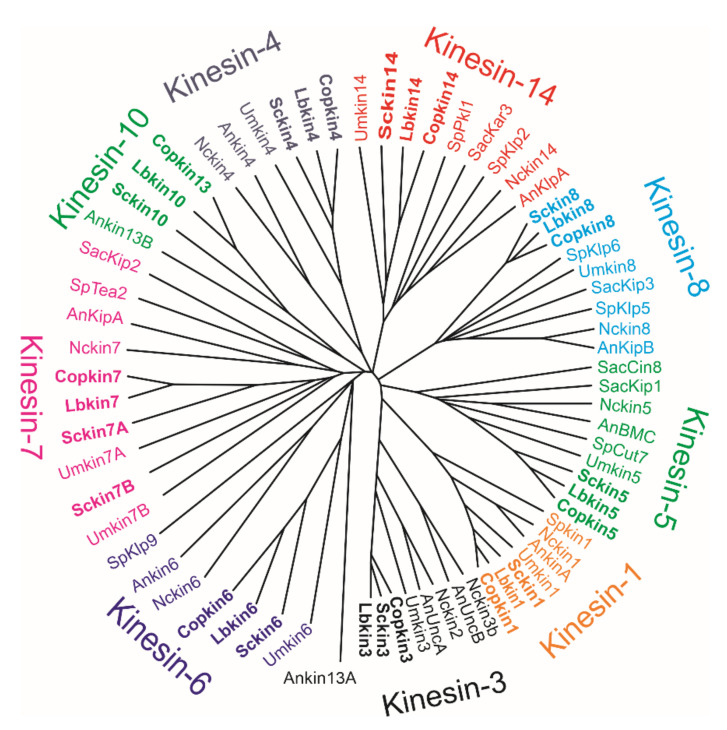
The tree is based on the alignment of the kinesin motor domains of the filamentous basidiomycetes *Schizophyllum commune*, *Coprinopsis cinerea* and *Laccaria bicolor*, basidiomycete smut *Ustilago maydis*, filamentous ascomycetes *Aspergillus nidulans* and *Neurospora crassa*, and yeasts *Saccharomyces cerevisiae and Schizosaccharomyces cerevisiae*. The GeneBank Accession or PID numbers of the kinesin proteins used for motor domain alignments are shown in Table 1. The basidiomycetes group into nine kinesin families known and identified in filamentous ascomycetes and yeasts. The filamentous basidiomycetes (Agaricomycetes) shown in bold are closely related in each kinesin family. The motor domains were aligned with Clustal Omega < Multiple Sequence Alignment < EMBL-EBI, and the tree was drawn with Phylogeny.fr.Drawtree 3.67.

**Figure 3 jof-08-00294-f003:**
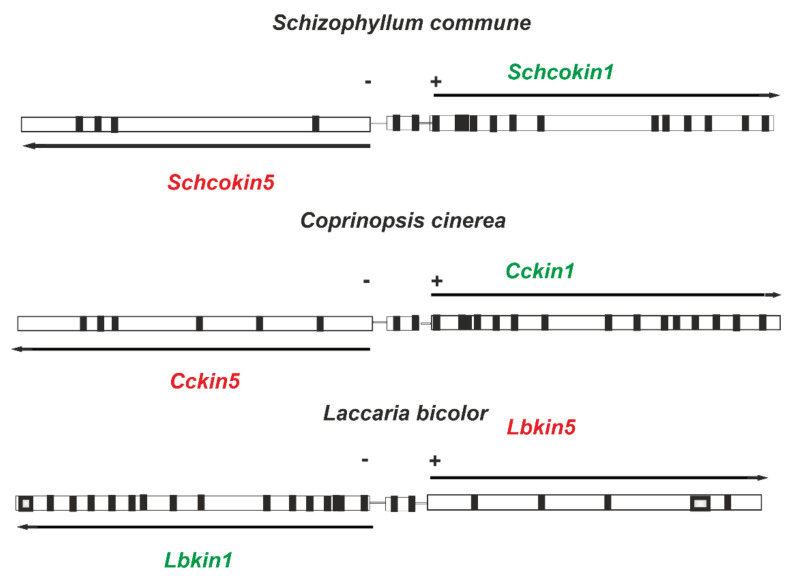
In the *Schizophyllum commune* genome (*Schizophyllum commune* H4-8 v3.0) the genes *Schco1* (encoding PID 2605236) and *Schco5* (encoding PID2605230) are tightly linked. In the approximately 1000 bp intergenic region, a small gene of about 470 bp is located (PID 2605233) with two introns and a hypothetical function. Similar intergenic organizations between genes *kin1* and *kin5* occur, also in the *Coprinopsis cinerea* genome (https://mycocosm.jgi.doe.gov/Copci1/Copci1.home.html; accessed on 7 February 2022, kin1 PID 3681 and kin5 PID 3679) and *Laccaria bicolor* genome (https://mycocosm.jgi.doe.gov/Lacbi2/Lacbi2.home.html; accessed on 7 February 2022, kin1 PID305999 and kin5 PID322200).

**Table 1 jof-08-00294-t001:** The GeneBank Accession or PID and reference numbers of kinesin proteins discussed in the text and presented in the tree based on the alignment of kinesin motor domains (Figure 2). Sac *Saccharomyces cerevisiae*, Sp *Schizosaccaromyces pombe*, An *Aspergillus nidulans*, Nc *Neurospora crassa*, Cop *Coprinopsis cinerea*, Lb *Laccaria bicolor*, Schco *Schizophyllum commune* and Um *Ustilago maydis*.

GeneBank Accession	References	GeneBank Assecion Number	References
SacCin8 NP_010853	[16,18,42,43,44]	Copkin1 XP_001833251.1	
SacKip1 NP_009490.1	[16,18,42,45]	Copkin3 XP_001841223.1	
Sac Kip2 NP_015170.1	[16,18,46,47,48]	Copkin4 XP_001833035.1	
SacKip3 NP_011299.1	[16,18,49]	Copkin5 XP_001833249.1	
Sack Kar3 NP_015467.1	[16,18,44,50]	Copkin6 XP_001828624.1	
		Copkin7 XP_001835410.1	
SpKlp3 CAB75775.1	[17,18]	Copkin8 XP_001836899.1	
SpCut7 CAA94636.1	[17,18,51,52]	Copkin13 XP_001832695.1	
Spkin6 Klp9 CAA21179.2	[17,18,53]	Copkin14 XP_001831857.1	
SpkinTea2 CAA22353.1	[17,18,54]		
SpkinKlp5 CAB10160.1	[17,18,55,56]	Lbkin1 XP_001875399.1	
SpkinKlp6 CAA20063.2	[17,18,55]	Lbkin3 XP_001884395.1	
SpkinKlp2 CAB65811.1	[17,18,57]	Lbkin4 XP_001875444.1	
SpkinPkl1 CAB16597.1	[17,18,57]	Lbkin5 XP_001874862.1	
		Lbkin6 XP_001873613.1	
AnkinA XP_662947.1	[18,19,58,59,60]	Lbkin7 XP_001875606.1	
AnUncA XP_680816.1	[19,61]	Lbkin8 XP_001874733.1	
AnUncB XP_664467.1	[19,61]	LbkinTPR XP_001881344.1	
Ankin4 XP_664479.1	[19]	Lbkin14 PID985042	
AnBMC XP_660967.1	[19,62,63,64]		
Ankin6 XP_660728.1		Schcokin1 ACG58879.1	
AnKipA XP_681555.1	[19,65,66,67]	Schcokin3 XP_003036969.1	
AnKipB XP_662117.1	[19,68]	Schcokin4 EU860363	
Ankin13A XP_661574.1		Schcokin5 EU850808.1	[14]
Ankin13B XP_661325.1	[19]	Schcokin6 PID2564434	
AnKlpA XP_663944.1	[18,19,69,70]	Schcokin7A XP_003026378.1	
		Schcokin7B XP_003025887.1	
Nckin1 KHC XP_964432.2	[18,19,50,51,52,53,54,55,71,72,73,74,75,76,77,78,79]	Schcokin8 XP_003036868.1	
Nckin2 XP_960661.2	[19,80]	Schcokin10 XP_003028679.1	
Nckin3b XP_961491.1	[19]	Schcokin14 XP_003032452.1	[14]
Nckin4 XP_963673.1	[19]		
Nckin5 XP_964753.1	[19]	Umkin1 XP_760365.1	[18,20,74,81]
Nckin6 XP_961843.1	[19]	Umkin3 XP_762398.1	[20,82,83,84,85,86,87,88,89,90,91]
Nckin7 XP_964051.1	[19]	Umkin4 XP_759304.1	[18,20,74]
Nckin8 XP_960006.2	[19]	Umkin5 XP_760872.1	[20,92]
Nckin14 XP_958282.1	[19]	Umkin6 XP_760874.1	[20]
		Umkin7A XP_760671.1	[20]
		Umkin7B XP_757043.1	[20]
		Umkin8 XP_757707.1	[20]
		Umkin14 XP_760654.1	[20]

No number no published work.

## Data Availability

https://mycocosm.jgi.doe.gov (accessed on 11 February 2022) for genome sequences of fungi and cited references. All data are publicly available.

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
