# Peer review of "Kinesin Motors in the Filamentous Basidiomycetes in Light of the Schizophyllum commune Genome"

_jof, 2022, doi:10.3390/jof8030294_

Round 1

Reviewer 1 Report

Dear Editor and author,

this paper is a wonderful review focusing on the Kinesin Motors in the Filamentous Basidiomycetes in light of the Schizophyllum commune genome, but it has to be check the follow comments and revised manuscript before accept.

  Introduction 1 please add the authority for fungal taxa on first time for all taxa in whole text;   2 add the taxonomic cllasification for the species Schizophyllum commune to authors;   3 "The number of kinesins is low in fungi, while humans have 45 known kinesins and 62 Arabidopsis has 61 known kinesins [21]", please give the dataset for  the number of kinesins in fungi, and detailed which  fungal taxa are the kinesins having;   Figures:   1 "Figure 1. Schizophyllum commune kinesin genes and proteins"; the letters are blurred, please polish it and make it more clear; 2 "Figure 2. The tree is based on the alignment of the kinesin motor domains of filamentous basidiomycetes Schizophyllum commune"; the fig. 2 is so careless, and it has to recheck, eg. some words are cut on the head and foot, check it.     Table 1 1 please add the references for a new line for the GeneBank Accession or PID numbers of kinesin proteins you employed;     References there are many format error, please check and revise them.  

This manuscript is focusing on a review about the Kinesin Motors in the Filamentous Basidiomycetes in light of 2 the Schizophyllum commune genome, which is very interesting to author. But it can be accepted after minor revision.

The comments follow the revised manuscript.

Kind Regards,

Author Response

Respond to Reviewer 1

(1) please add the authority for fungal taxa on first time for all taxa in whole text;

The fungal taxa, authority of the species is added on the first time the species is mentioned. In Introduction rows 57-60, rows 159-160 in comparison of kinesin genes, the additions are marked by red

(2) add the taxonomic cllasification for the species Schizophyllum commune to authors;   3 "The number of kinesins is low in fungi, while humans have 45 known kinesins and 62 Arabidopsis has 61 known kinesins [21]", please give the dataset for the number of kinesins in fungi, and detailed which fungal taxa are the kinesins having;  

The names of the species and exact references are added to text dealing with kinesin numbers in fungi. Ref. 18 fits better to yeasts, since all the yeast kinesins are given, but only two kinesins for filamentous ascomycete Aspergillus nidulans, in revised manuscript rows 57-60.

Figures:   1 "Figure 1. Schizophyllum commune kinesin genes and proteins"; the letters are blurred, please polish it and make it more clear; 2 "Figure 2. The tree is based on the alignment of the kinesin motor domains of filamentous basidiomycetes Schizophyllum commune"; the fig. 2 is so careless, and it has to recheck, eg. some words are cut on the head and foot, check it.

Figure 1 and 2 have been improved. Hopecully the letters in Figure 1A are clearer and the Tree in Figure 2 is acceptable. -I have asked the Editor of Journal of Fungi to insert the revised Figures 1 and 2 in the revised manuscript, my own or my computer abilities were not enough. I really wish that they look better now.

Table 1 please add the references for a new line for the GeneBank Accession or PID numbers of kinesin proteins you employed;  

The reference numbers of the most central publications have been added to Table1, which now visualizes very well how little/almost nothing is published about kinesins in filamentous basidiomycetes. Thank you for this suggestion.

References there are many format errors, please check and revise them.

Thank you for the comments on the manuscript and those especially that show the errors in the list of References. 

Reviewer 2 Report

This review is on kinesin motors in the Filamentous Basidiomycetes especially Schizophyllum commune. It also provides a detailed introduction of the kinesin motors especially those in yeasts and other filamentous fungi including Ustilago maydis, Aspergillus nidulans and Neurospora crassa. Overall, I am very impressed by this comprehensive review, which should be of great help to the fungal field. While I strongly support its publication, I would like to include several minor suggestions and questions for the author to consider during the revision.

  1. Line 40, what is “recoding”?
  2. Line 71, add references after the sentence.
  3. Line 120, add a reference after the sentence about the light chains.
  4. Line 134, add references about the release from autoinhibition of kinesin-1.
  5. Line 204, add references.
  6. Line 234, the nuclear distribution phenotype of the kinesin-1 mutant is possibly related to a defect in dynein localization to the microtubule plus ends, which was found in Aspergillus nidulans (Zhang et al., 2003 Mol Biol Cell) and Ustilago maydis (Lenz et al., 2006 EMBO).
  7. When you talked about kinesin-1, it would also be good to mention that kinesin-1 is important for transporting secretary vesicles, a function almost essential for fungal growth in the absence of myosin V, as these motors cooperate (Schuster et al., 2012 EMBO; Penalva et al., 2017 Mol Biol Cell).
  8. Line 293-303, you can add more papers on cargoes that hitchhike on the motile early endosomes to get distributed inside hyphae (reviewed by Salogiannis and Reck-Peterson 2017 Trends Cell Biol).
  9. Line 340, the reference needs to be replaced by a number.
  10. Line 377, on kinesin-14’s role in spindle formation, Prigozhina et al., Mol Boil Cell 2001 may be included.
  11. Line 499, change “KinA” to “KipA”. On kinesin-7’s role in microtubule dynamics, it would be better to mention their function in the plus-end accumulation of the microtubule plus end-tracking protein CLIP170 homologs (that controls microtubule dynamics) in S. cerevisiae, S. pombe and A. nidulans (Carvalho et al., 2004 Dev Cell; Busch et al., 2004 Dev Cell; Efimov et al., 2006 Mol Cell Biol). In addition, Kip2 also plays a role in transporting dynein to the plus end (Carvalho et al., 2004 Dev Cell; Roberts et al., eLife 2014).
  12. Line 615, it would be better to change “spindle elongation” to “spindle orientation”, which would more accurately reflect the main points of Ref 138.

Author Response

Respond to Reviewer 2

  1. Line 40, what is “recoding”? Corrected with the ”visualization” line 40
  2. Line 71, add references after the sentence. References and species are added, lines 71-74.
  3. Line 120, add a reference after the sentence about the light chains. Reference 38 added . The reference is a good review from 2009, line 124
  4. Line 134, add references about the release from autoinhibition of kinesin-1. Ref.38, see above
  5. Line 204, add references.References 33-37 are added, which include information about kinesin motors both from animals and fungi., line 209
  6. Line 234, the nuclear distribution phenotype of the kinesin-1 mutant is possibly related to a defect in dynein localization to the microtubule plus ends, which was found in Aspergillus nidulans (Zhang et al., 2003 Mol Biol Cell) and Ustilago maydis (Lenz et al., 2006 EMBO). -New text and reference 61 added, lines245-248 marked with red in the revised manuscript.
  7.  When you talked about kinesin-1, it would also be  good to mention that kinesin-1 is important for transporting secretary vesicles, a function almost essential for fungal growth in the absence of myosin V, as these motors cooperate (Schuster et al., 2012 EMBO; Penalva et al., 2017 Mol Biol Cell). -New text and references 58, 60, 61 added, ref 57 was already included in the original manuscript, lines 232-238 marked with red in revised manuscript.
  8. Line 293-303, you can add more papers on cargoes that hitchhike on the motile early endosomes to get distributed inside hyphae (reviewed by Salogiannis and Reck-Peterson 2017 Trends Cell Biol).-Reference and new text included, lines 311-313 in red in revised manuscript
  9. Line 340, the reference needs to be replaced by a number,-ref. 80, line 323
  10. Line 377, on kinesin-14’s role in spindle formation, Prigozhina et al., Mol Boil Cell 2001 may be included.- Included ref. 102, line 391, fits best at this site.
  11. Line 499, change “KinA” to “KipA”. corrected .On kinesin-7’s role in microtubule dynamics, it would be better to mention their function in the plus-end accumulation of the microtubule plus end-tracking protein CLIP170 homologs (that controls microtubule dynamics) in S. cerevisiae, S. pombe and A. nidulans (Carvalho et al., 2004 Dev Cell; Busch et al., 2004 Dev Cell; Efimov et al., 2006 Mol Cell Biol). In addition, Kip2 also plays a role in transporting dynein to the plus end (Carvalho et al., 2004 Dev Cell; Roberts et al., eLife 2014). -The function of yeast Kip2 and Bik1 proteins is discussed as well as ClipA association with KinA ,KipA and dynein in Aspergillus. The references 58, 127, 128, 129 are included, lines 525-537, marked by red in the revised manuscript.
  12. Line 615, it would be better to change “spindle elongation” to “spindle orientation”, which would more accurately reflect the main points of Ref 138.- Corrected, now reference 146, lane 641.
  13. I thank the Reviewer for the useful references. I also hope that my interpretation about the contents of the references added to the manuscript are acceptable.